# Hierarchy of TGFβ/SMAD, Hippo/YAP/TAZ, and Wnt/β-catenin signaling in melanoma phenotype switching

Fabiana Lüönd[1,*], Martin Pirkl[2,3,*] , Mizue Hisano[1], Vincenzo Prestigiacomo[1], Ravi KR Kalathur[1], Niko Beerenwinkel[2,3] , Gerhard Christofori[1] 

In melanoma, a switch from a proliferative melanocytic to an invasive mesenchymal phenotype is based on dramatic transcriptional reprogramming which involves complex interactions between a variety of signaling pathways and their downstream transcriptional regulators. TGFβ/SMAD, Hippo/YAP/TAZ, and Wnt/β-catenin signaling pathways are major inducers of transcriptional reprogramming and converge at several levels. Here, we report that TGFβ/SMAD, YAP/TAZ, and β-catenin are all required for a proliferative-to-invasive phenotype switch. Loss and gain of function experimentation, global gene expression analysis, and computational nested effects models revealed the hierarchy between these signaling pathways and identified shared target genes. SMAD-mediated transcription at the top of the hierarchy leads to the activation of YAP/TAZ and of β-catenin, with YAP/TAZ governing an essential subprogram of TGFβ-induced phenotype switching. Wnt/β-catenin signaling is situated further downstream and exerts a dual role: it promotes the proliferative, differentiated melanoma cell phenotype and it is essential but not sufficient for SMAD or YAP/TAZ–induced phenotype switching. The results identify epistatic interactions among the signaling pathways underlying melanoma phenotype switching and highlight the priorities in targets for melanoma therapy.

## Introduction

Melanoma arising by transformation of melanocytes is one of the most aggressive and deadliest cancers. To describe the malignant progression of melanoma from benign horizontal growth to invasive cancer and metastasis formation, a melanoma phenotype switching model has been proposed. This model pictures that melanoma progression and metastasis are driven by a continuous switching between two cellular phenotypes: a differentiated, "proliferative" phenotype characterized by high expression of neural crest and melanocyte markers, such as microphthalmia-associated transcription factor (MITF), and a dedifferentiated, "invasive" phenotype characterized by low expression of melanocyte markers and high expression of mesenchymal cell markers (Hoek et al, 2006, 2008; Hoek & Goding, 2010; Widmer et al, 2012). Besides driving invasion and metastasis, the invasive, mesenchymal phenotype has been associated with resistance to targeted therapies and immunotherapies (Zipser et al, 2011; Landsberg et al, 2012; Muller et al, 2014; Tirosh et al, 2016; Boshuizen et al, 2018).

The melanoma phenotype switch from a proliferative to an invasive phenotype involves reversible epigenetic changes and massive transcriptional reprogramming, analogous to an epithelial–mesenchymal transition (EMT) observed in carcinomas (Caramel et al, 2013; Vandamme & Berx, 2014; Schlegel et al, 2015; Falletta et al, 2017; Wouters et al, 2020). Well-studied in embryonic development and carcinomas, EMT can be induced by activation of a variety of signaling pathways. There is extensive crosstalk between these signaling pathways and their downstream transcriptional regulators which jointly control the transcriptional reprogramming during EMT (Lamouille et al, 2014). Of note, EMT is not a binary switch but covers a continuum of intermediate states under the control of distinct and hierarchical regulatory networks (Huang et al, 2013; Zhang et al, 2014; Jolly et al, 2016; Pastushenko et al, 2018; Meyer-Schaller et al, 2019; Yang et al, 2020). While the existence of intermediate states during melanoma phenotype switching has recently been reported (Tirosh et al, 2016; Tsoi et al, 2018; Rambow et al, 2019; Tuncer et al, 2019; Wouters et al, 2020), less is known about the crosstalk and hierarchy of signaling pathways inducing a proliferative-to-invasive phenotype switch.

The TGFβ/SMAD2/3, Hippo/YAP/TAZ, and Wnt/β-catenin pathways are important triggers of EMT during embryonic development and in carcinomas (Lei et al, 2008; Xu et al, 2009; Varelas et al, 2010; Lamar et al, 2012; Massague, 2012; Lamouille et al, 2014; Zanconato et al, 2016). Whereas TGFβ/SMAD2/3 and Hippo/YAP/TAZ have also been shown to be strong inducers of a proliferative-to-invasive phenotype switch and to promote melanoma progression and metastasis (Perrot et al, 2013; Nallet-Staub et al, 2014; Schlegel et al, 2015; Verfaillie et al, 2015), the role of Wnt/β-catenin signaling in

---

[1]Department of Biomedicine, University of Basel, Basel, Switzerland   [2]Department of Biosystems Science and Engineering, ETH Zurich, Basel, Switzerland   [3]SIB Swiss Institute of Bioinformatics, Basel, Switzerland

Correspondence: gerhard.christofori@unibas.ch
*Fabiana Lüönd and Martin Pirkl contributed equally to this work

melanoma progression is obscured by conflicting experimental results (Widlund et al, 2002; Chien et al, 2009; Arozarena et al, 2011; Damsky et al, 2011; Eichhoff et al, 2011; Kovacs et al, 2016). On the other hand, there is extensive crosstalk between canonical TGFβ, Hippo, and Wnt signaling pathways. For example, SMAD2/3, YAP/TAZ, and β-catenin impinge on common target genes, resulting in context-dependent transcriptional changes, which might explain the conflicting results regarding β-catenin and melanoma progression (Attisano & Wrana, 2013, Piersma et al, 2015a, 2015b). Hence, the hierarchy and the order of events executed by these signaling pathways and their transcriptional regulators have remained elusive.

Nested effects models (NEMs) provide a statistical framework for computationally inferring the hierarchy of signaling pathways (Markowetz et al, 2007). The signaling genes (called S-genes) of the pathway are perturbed in different experiments and the expression of the effect reporters (called E-genes) is measured. The differential expression profiles of the perturbations are compared and the hierarchy can be resolved. For example, if S-gene A is upstream of S-gene B, the set of E-genes changing expression during a perturbation of B are a subset of the E-genes changing expression during a perturbation of A because the perturbation of A is propagated to also perturb B.

Using a systems biology approach based on in vitro perturbation and NEM computation, we investigated the functional relevance and hierarchy of TGFβ/SMAD, Hippo/YAP/TAZ, and Wnt/β-catenin signaling in a proliferative-to-invasive phenotype switch in human melanoma cell lines. We found that SMAD, YAP/TAZ, and β-catenin–mediated transcriptional control are all required for a proliferative-to-invasive phenotype switch of differentiated melanoma cells. TGFβ/SMAD signaling appears to rank on top of the hierarchy with Hippo/YAP/TAZ located downstream, yet playing a critical part in the induction of the invasive phenotype. β-catenin acts even further downstream and is required yet not sufficient to induce melanoma phenotype switching. In fact, β-catenin exerts a dual role by supporting the proliferative, differentiated phenotype of melanoma cells and by supporting the TGFβ/SMAD– and Hippo/YAP/TAZ–induced proliferative-to-invasive phenotype switch.

# Results

### The distinct functions of TGFβ/SMAD, Hippo/YAP/TAZ, and Wnt/β-catenin signaling

To address the role of canonical SMAD, YAP/TAZ and β-catenin transcriptional activity in melanoma phenotype switching, we first assessed whether their activation is sufficient to induce a proliferative-to-invasive phenotype switch. To this end, we treated proliferative M000921 and M010817 patient-derived melanoma cells (Hoek et al, 2006) with either recombinant human TGFβ or with recombinant murine Wnt-3a, or we knocked down LATS1 and LATS2 to induce the transcriptional activity of YAP/TAZ. Compared to their individual siRNA-mediated depletion, the combined depletion of LATS1 and LATS2 (referred to as siLATS1/2) led to the most pronounced YAP/TAZ transcriptional activity marked by the up-regulated expression of CYR61, CTGF, SERPINE1, and ANKRD1 (Fig S1A and B). As previously described (Schlegel et al, 2015), TGFβ

treatment induced an invasive phenotype in both cell lines, marked by a more prominent mesenchymal morphology (Figs 1A and S2A) and a strong down-regulation of the expression of the melanocyte marker genes MITF and MLANA as well as the concomitant induction of the expression of the mesenchymal marker genes FN1 and CDH2 and the EMT-inducing transcription factors SNAI1 and ZEB1 (Figs 1B and S2B–D). Similarly, knockdown of LATS1/2 and thus activation of YAP/TAZ activity led to a strong down-regulation of melanocyte markers, but only a mild up-regulation of mesenchymal markers and EMT-inducing transcription factors in M000921 and M010817 cells. Yet, activation of YAP/TAZ led to a milder induction of the invasive phenotype than TGFβ treatment (Figs 1A and B and S2A and B). In contrast, Wnt-3a treatment did neither affect cell morphology nor the expression of mesenchymal marker genes. It rather led to a weak increase in melanocyte marker expression, particularly MITF, suggesting that canonical Wnt/β-catenin signaling rather promotes the proliferative, differentiated phenotype (Figs 1A and B and S2A and B). Interestingly, TGFβ-treatment induced the expression of well-established canonical TGFβ/SMAD, Hippo/YAP/TAZ, and Wnt/β-catenin target genes, suggesting an activation of all three transcriptional pathways. In contrast, activation of YAP/TAZ by siRNA-mediated depletion of LATS1/2 did not affect TGFβ target genes and EMT transcription factors, yet substantially induced the expression of its canonical target genes and slightly repressed canonical Wnt target genes (Figs 1B and S2B–D). Finally, stimulation of the cells with Wnt-3a induced the expression of the canonical Wnt target genes AXIN2, NOTUM, NKD1, and CTLA4, yet did not affect the expression canonical TGFβ and YAP/TAZ target genes (Figs 1B and S2B). Whereas Wnt-3a treatment induced an efficient translocation of β-catenin to the nucleus of the proliferative melanoma cells, TGFβ treatment also relocalized β-catenin to the nucleus, although to a lesser extent (Fig S3A and B). Finally, the same changes in gene expression were also observed in cells that have been treated with TGFβ or Wnt-3a in the absence of any siCtrl transfections (Fig S4A and B).

To functionally validate the proliferative-to-invasive phenotype switch of the melanoma cells, the M010817 cells were cultured in 3D Matrigel culture conditions and subjected to modified Boyden chamber cell migration assays. Treatment of the cells for 5 d with TGFβ and with siLATS1/2, but not with Wnt-3a, resulted in a phenotype switch in Matrigel culture, with elongated cell morphology and invasive growth (Fig 1C). In contrast, only the treatment with TGFβ induced an increase in cell migration in modified Boyden chamber assays (Fig 1D).

Together, these data suggest that TGFβ acts as a strong inducer of proliferative-to-invasive phenotype switching and leads to the activation of SMAD, YAP/TAZ, and β-catenin transcriptional activity, with YAP/TAZ potentially governing a transcriptional subprogram thereof, whereas canonical Wnt/β-catenin signaling rather appears to promote the proliferative cell phenotype.

### SMAD, TAZ, and β-catenin transcriptional activities are required for proliferative-to-invasive phenotype switching

To assess whether SMAD, YAP/TAZ, and β-catenin–dependent transcriptional activities are functionally required for a proliferative-to-invasive phenotype switch, we ablated the expression of SMAD4, YAP, TAZ, and β-catenin (CTNNB1) by siRNA-mediated knockdown in

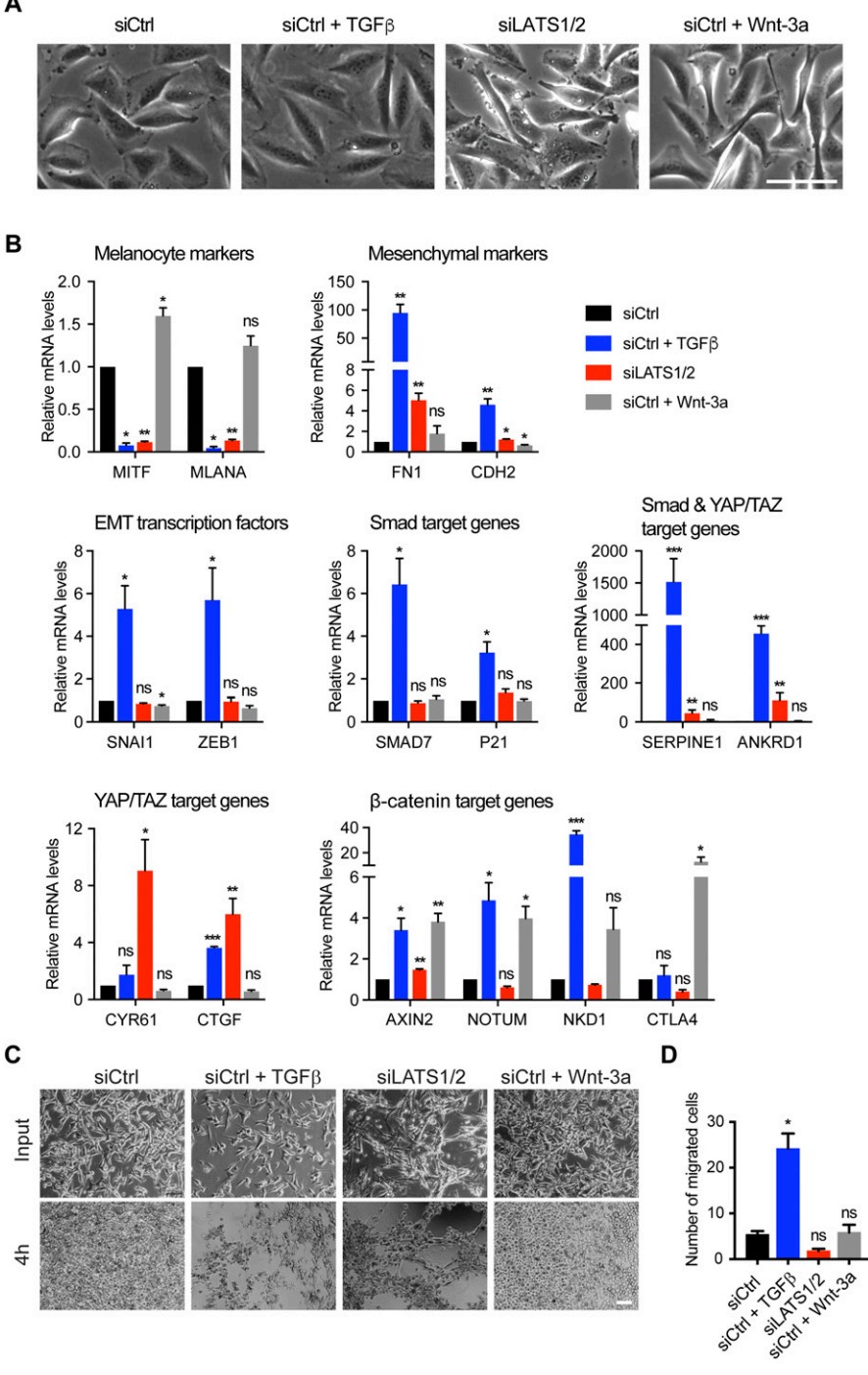

**Figure 1. TGFβ/SMAD and YAP/TAZ induce an invasive cell phenotype, whereas Wnt/β-catenin promotes a proliferative cell phenotype in proliferative-type melanoma cells.**
**(A)** Representative phase-contrast images of proliferative-type M000921 cells treated with TGFβ, siLATS1/2, or Wnt-3a for 2 d. Scale bar, 100 μm.
**(B)** Quantitative RT-PCR analysis of the expression of melanocyte marker genes (*MITF* and *MLANA*), mesenchymal marker genes (*FN1* and *CDH2*), EMT transcription factors (*SNAI1* and *ZEB1*) and SMAD (*SMAD7* and *P21*), SMAD/YAP/TAZ (*SERPINE1* and *ANKRD1*), YAP/TAZ (*CYR61* and *CTGF*), and β-catenin target genes (*AXIN2*, *NOTUM*, and *NKD1*) in proliferative-type M000921 cells treated with siControl (siCtrl), siCtrl + TGFβ, siLATS1/2, or siCtrl + Wnt-3a for 2 d. Mean + SEM of n = 3 replicates are shown *P < 0.05; **P < 0.01; ***P < 0.001; ratio-paired *t* test. **(C)** Invasive growth in 3D Matrigel culture. Proliferative-type M010817 cells were transfected with siCtrl or LATS1/2 or treated with TGFβ or Wnt-3a for 5 d as indicated and subsequently cultured in diluted 3D Matrigel. Representative pictures were taken by phase-contrast microscopy after seeding and 4 h thereafter, revealing the morphological changes and invasive growth of TGFβ and siLATS1/2-treated cells in Matrigel. Scale bar, 100 μm. **(D)** Cell migration assay. Proliferative-type M010817 melanoma cells were transfected with siCtrl or siLATS1/2 or treated with TGFβ or Wnt-3a for 5 d and subsequently seeded in modified Boyden Chamber culture insets with 10% FBS as chemoattractant in the bottom well. Cells that have migrated after treatment with siCtrl, siLATS1/2, TGFβ, or Wnt-3a were quantified after 20 h. *P < 0.05; ratio-paired *t* test.

combination with TGFβ treatment or upon knockdown of LATS1/2. In both proliferative phenotype melanoma cell lines, knockdown of SMAD4, TAZ, and β-catenin, but not of YAP, significantly counteracted the TGFβ-induced down-regulation of the expression of the melanocyte markers *MITF* and *MLANA* (Figs 2A and S5A). As expected, SMAD4 knockdown prevented the TGFβ-induced up-regulation of mesenchymal markers, EMT-transcription factors, and canonical TGFβ and TAZ target genes. Similar but less pronounced effects were observed upon

TAZ or β-catenin knockdown (Figs 2A and S5A). Together, these data suggest that SMAD2/3, TAZ, and β-catenin are essential for a TGFβ-induced phenotype switch. Interestingly, in both cell lines, the phenotype switch and expression of YAP/TAZ target genes induced by LATS1/2 knockdown was almost completely abrogated upon loss of TAZ, whereas loss of YAP had no or only week effects in M000921 and M010817 cells, respectively (Figs 2B and S5B). Loss of β-catenin significantly counteracted the siLATS1/2–induced phenotype switch, yet

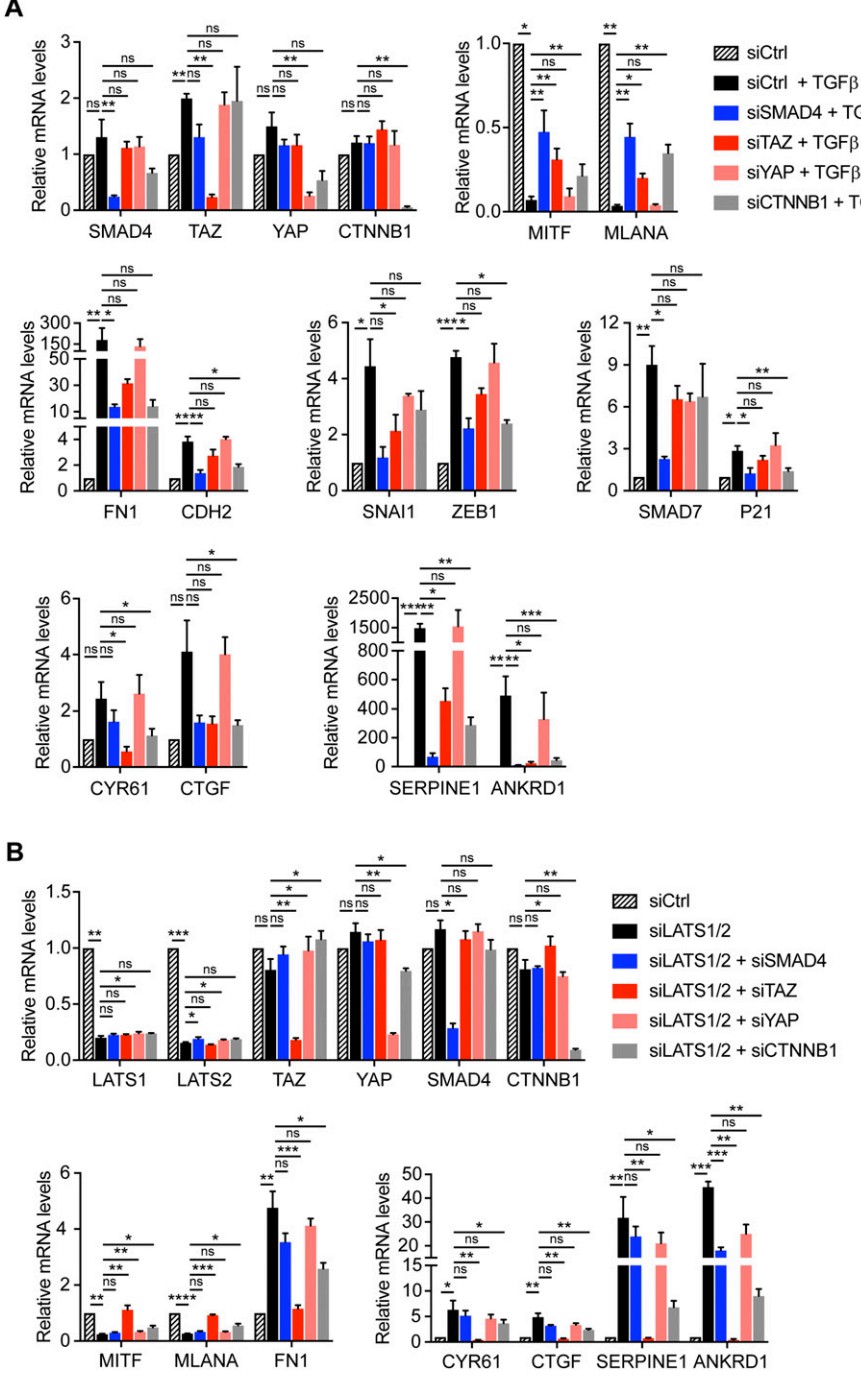

**Figure 2. SMAD4, TAZ, and β-catenin are required for a proliferative-to-invasive phenotype switch in M000921 cells.**

**(A)** Quantitative RT-PCR analysis to assess knockdown efficiencies as well as expression of melanocyte marker genes (*MITF* and *MLANA*), mesenchymal marker genes (*FN1* and *CDH2*), epithelial–mesenchymal transition transcription factors (*SNAI1* and *ZEB1*), and SMAD (*SMAD7* and *P21*), YAP/TAZ (*CYR61* and *CTGF*), and SMAD/YAP/TAZ target genes (*SERPINE1*, *ANKRD1*) upon knockdown of SMAD4, TAZ, YAP, and β-catenin (*CTNNB1*) during 2 d of a TGFβ-induced phenotype switch. **(B)** Quantitative RT-PCR analysis to assess knockdown efficiencies as well as expression of melanocyte marker genes (*MITF* and *MLANA*), mesenchymal marker genes (*FN1*), and YAP/TAZ target genes (*CYR61*, *CTGF*, *SERPINE1*, and *ANKRD1*) upon knockdown of SMAD4, TAZ, YAP, and β-catenin (*CTNNB1*) during 2 d of an siLATS1/2-induced phenotype switch. Mean + SEM of n = 3 replicates are shown. *$P < 0.05$; **$P < 0.01$; ***$P < 0.001$; ratio-paired *t* test.

not as pronounced as knockdown of TAZ. In contrast, knockdown of SMAD4 had none to very weak inhibitory effects on the siLATS1/2–induced phenotype switch, which would be consistent with TAZ being downstream of SMAD activity during a TGFβ-induced phenotype switch (Figs 2B and S5B). Knockdown efficiency in these experiments was determined by immunoblotting (Fig S5C).

In the absence of any TGFβ stimulation, siRNA-mediated ablation of SMAD4, TAZ, YAP, and β-catenin (*CTNNB1*) did not induce the expression of mesenchymal markers, yet rather increased the expression of the melanocytic marker genes *MITF* and *MLANA* (Fig S6A and B). Of note, in unstimulated cells, knockdown of TAZ led to down-regulation of its canonical target genes and a very slight induction of melanocyte marker genes and down-regulation of *FN1* and *CDH2* expression suggesting that low baseline TAZ activity is not sufficient to induce an invasive phenotype but might prime cells towards a phenotype switch (Fig S6A and B). The specific ablation of β-catenin was further confirmed by the use of different pools of siRNA sequences (Fig S7A and B).

Besides its transcriptional activity in canonical Wnt signaling, β-catenin exerts important functions in cadherin-mediated cell adhesion (Nelson & Nusse, 2004). As N-cadherin (*CDH2*) is up-regulated in the invasive cell state, we hypothesized that β-catenin might be required for induction of a mesenchymal phenotype because of its role in N-cadherin–mediated cell adhesion. Indeed, N-cadherin and β-catenin co-localized at the cell membrane and upon knockdown of N-cadherin, β-catenin levels were dramatically reduced (Fig S8A and B). Moreover, co-immunoprecipitation of β-catenin and N-cadherin confirmed the interaction of these proteins (Fig S9A). However, loss of membrane-bound β-catenin did not counteract a TGFβ or siLATS1/2-induced proliferative-to-invasive phenotype switch (Fig S9B and C). Interestingly, the siRNA-mediated ablation of N-cadherin (*CDH2*) expression only moderately increased the expression Wnt/β-catenin target genes either in the absence of any additional stimulation or upon treatment with siLATS1/2 or TGFβ (Fig S10A and B). These data suggest that β-catenin's transcriptional activity rather than its cell adhesion function is required for a proliferative-to-invasive phenotype switch.

Finally, to further assess the functional contribution of Wnt/β-catenin transcriptional output to melanoma cell phenotype switching we ablated the expression of TCF4 and LEF1, the TCF family members predominantly expressed in melanoma cells. siRNA-mediated ablation of TCF4 or LEF1 and mostly the combination of both resulted into the reduced expression of some but not all Wnt3a-induced Wnt/β-catenin target genes and of siLATS1/2-induced YAP/TAZ target genes, yet not in substantial changes in the expression of melanocytic and mesenchymal marker genes (Fig S11A–D).

Thus, even though canonical Wnt/β-catenin signaling rather promotes the proliferative, differentiated phenotype, β-catenin transcriptional activity is required for the induction of the invasive phenotype, suggesting that β-catenin might have a dual role in melanoma progression. However, its contribution to the expression of genes induced by TGFβ or YAP/TAZ seems to vary, potentially explaining its auxiliary contribution to melanoma cell phenotype switching.

### Global effects of TGFβ/SMAD, Hippo/YAP/TAZ, and Wnt/β-catenin signaling on melanoma cell phenotypes

To investigate the crosstalk and hierarchy of SMAD, YAP/TAZ, and β-catenin signaling on a global level, we performed RNA sequencing analysis of M000921 and M010817 cells subjected to all single and combinatorial double and triple treatments with TGFβ, siLATS1/2, and Wnt-3a.

Principal component analysis revealed that the strongest effects observed were due to the differences between the two patient-derived cell lines (Fig S12A). However, upon removal of the cell line effect, four clusters corresponding to the treatments could be observed (Fig S12B). Interestingly, consistent with the functional data presented above, Wnt-3a treatment had little global effects and clustered together with the control or, in combination treatments, together with the respective single or double treatments. As we had previously obtained very consistent results with both cell lines and were interested in robust transcriptional activity without any cell line–specific effects, we combined the data from both cell

lines for further analysis. Overall, Wnt-3a treatment alone triggered few transcriptional changes, whereas siLATS1/2 and particularly TGFβ treatment provoked dramatic transcriptional reprogramming, as expected (Fig 3A and B). Combination of two or three treatments, including the combination of siLATS1/2 and TGFβ, did not cause more dramatic changes than the single treatments with either siLATS1/2 or TGFβ alone (Fig S13A–D).

Differential gene expression analysis revealed a large number of shared target genes between all three pathways, in particular between TGFβ and siLATS1/2 single treatments. Of note, the vast majority (82.5%) of these differentially regulated genes shared by TGFβ and siLATS1/2 were regulated in the same direction, consistent with the notion that both are inducers of the invasive phenotype (Figs 3C and D and S14). Similarly, ~60% of differentially expressed genes shared by Wnt-3a and TGFβ (62.7%) or Wnt-3a and siLATS1/2 (61.5%) were regulated in the same direction. Interestingly, for the genes that were regulated in opposite directions, for more than 95.8% (TGFβ versus Wnt-3a) and 98.7% (siLATS1/2 versus Wnt-3a) of genes, TGFβ and siLATS1/2 treatments were dominant over Wnt-3a treatment in the respective combinatorial treatments (Fig 3D and Table S1). Consistent with this, 49.3% of the differentially regulated genes shared by all three single treatments were regulated in the same direction, and both TGFβ and siLATS1/2 were dominant over Wnt-3a in the majority (59.1%) of genes that were regulated in opposite directions between different treatments (Fig 3D and Table S1).

For further in-depth analysis, we counted the genes following each of the $2^7 = 128$ possible differential expression patterns which may result from the seven treatment combinations when considering each gene as either up- or down-regulated. The two top ranking patterns represented genes differentially regulated in the same direction (up or down) by all seven treatment combinations (Fig 4A). These top patterns were followed by patterns in which Wnt3-a treatment regulated genes in the opposite direction of all other treatment combinations (Fig 4B). Thus, amongst commonly regulated genes, Wnt-3a treatment showed two predominant patterns: genes regulated in the same direction (Fig 4A, pattern 1) or opposite direction (Fig 4B, pattern 2) as compared against TGFβ or siLATS1/2 treatments. These data also confirmed that both TGFβ and siLATS1/2 were dominant over Wnt-3a in regulating the expression of most of the shared target genes. These results suggest that TGFβ and siLATS1/2 share a transcriptional program and that β-catenin is effective further downstream in the epistatic hierarchy.

To investigate the effect of TGFβ-, siLATS1/2-, and Wnt-3a–mediated signaling on melanoma phenotypes on a global level, we next performed gene set enrichment analysis using publicly available gene sets characterizing the proliferative or invasive melanoma cell state (Widmer et al, 2012; Verfaillie et al, 2015). As expected, the differentially regulated genes in Wnt-3a–treated cells were mildly, yet positively enriched for proliferative gene sets, whereas negatively enriched for invasive gene sets (Fig 5A and B). Conversely, in TGFβ and siLATS1/2-treated cells, the differentially expressed genes showed a strong positive enrichment for invasive gene sets and a negative enrichment for proliferative gene sets (Fig 5C–F). In the combinatorial treatments, TGFβ and siLATS1/2 again exerted dominant functions over Wnt-3a treatment and induced a strong enrichment for invasive gene sets (Fig S15A–H). Hierarchical

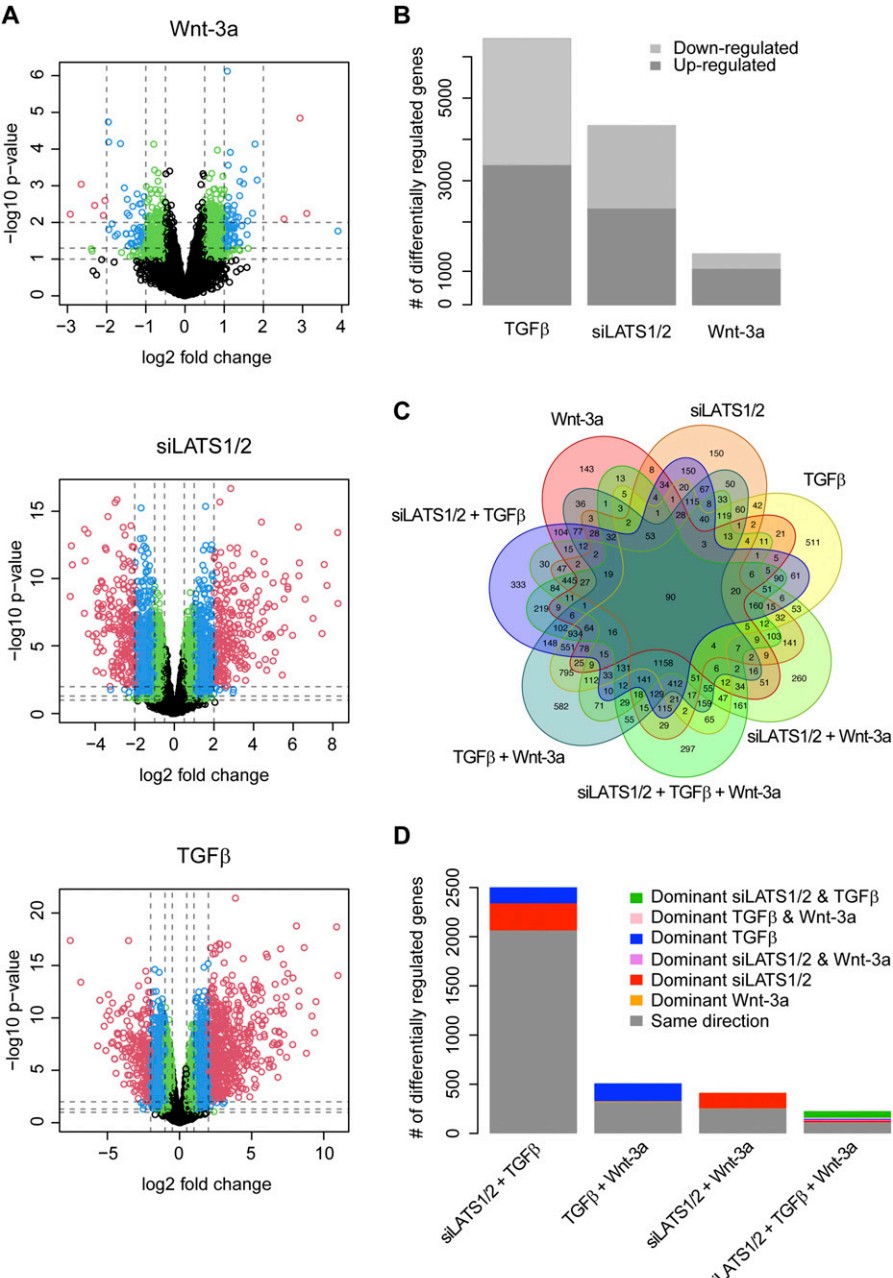

**Figure 3. Profound changes in gene expression upon 2 d of treatment with TGFβ, siLATS1/2, and Wnt-3a with a global dominance by TGFβ and siLATS1/2.**
**(A)** Volcano plots for differentially regulated genes of the single treatments (TGFβ, siLATS1/2, and Wnt-3a). The dashed vertical lines show the cutoffs for an absolute fold change of 0.5, 1, and 2. The horizontal lines show the cutoffs for a *P*-value of 0.1, 0.05, and 0.01. The colors correspond to genes with a fold change of greater than 0.5 and a *P*-value of less than 0.1 (green), greater than 1 and less than 0.05 (blue), and greater than 2 and less than 0.01 (red). **(B)** Bar plot showing the numbers of significantly regulated genes by the individual single treatments as well as the directionality of the regulation. **(C)** Venn diagram of the overlaps between the single and combined treatments. **(D)** Bar plot showing the numbers of commonly regulated genes between the individual treatments and whether these genes are regulated in the same direction, or if not, which treatments dominate over the others. Data of both cell lines was combined for this and all subsequent computational analysis. For the Venn diagram and bar plots an FDR cutoff of 10% was chosen.

---

clustering analysis using the Widmer and the Verfaillie gene sets confirmed that Wnt-3a treatment promoted the proliferative, differentiated melanoma cell phenotype, whereas TGFβ and siLATS1/2 treatment induced an invasive phenotype with a dominant effect over Wnt-3a treatment (Figs S16A and B and S17A and B).

### The hierarchy of TGFβ/SMAD, Hippo/YAP/TAZ, and Wnt/β-catenin signaling in melanoma phenotype switching

To delineate the hierarchy among TGFβ/SMAD, Hippo/YAP/TAZ, and Wnt/β-catenin signaling in the melanoma proliferation-to-invasive switch, we applied NEMs analysis. NEMs infer the hierarchy of signaling genes (S-genes) in a pathway from the global gene expression changes of the measured genes (effect reporters, E-genes) in a perturbation experiment compared with a control (Fig 6A). For the three S-genes TGFβ, siLATS1/2, and Wnt-3a, we considered all possible topologies and scored them against the observed effects, that is, the gene expression data. To assess the stability of the top-ranked NEM, we performed a bootstrap analysis, which showed a support of almost 100% for all edges (Fig 6B). No support for any additional edges was found. The optimal NEM predicts a linear hierarchy with TGFβ on top, siLATS1/2 downstream, and Wnt-3a at the bottom, downstream of both TGFβ, and siLATS1/2, thus confirming the experimental observations (Figs 1 and 2 and S2–S11) as

**Figure 4. Distinct gene expression patterns of Wnt-3a–regulated genes.**
**(A)** Gene expression pattern 1: Wnt-3a regulates genes in the same direction as the treatments with TGFβ or with siLATS1/2 and combinations thereof. **(B)** Gene expression pattern 2: Wnt-3a regulates genes in the opposite direction of the other treatments with TGFβ or with siLATS1/2 and combinations thereof. Common target genes of Wnt-3a, siLATS1/2, and TGFβ treatment with corrected P-values of less than 10% were ranked after their absolute log_2 fold change in the Wnt-3a single treatment and filtered for patterns 1 and 2. Log_2 fold changes of the top 30 ranked genes are shown.

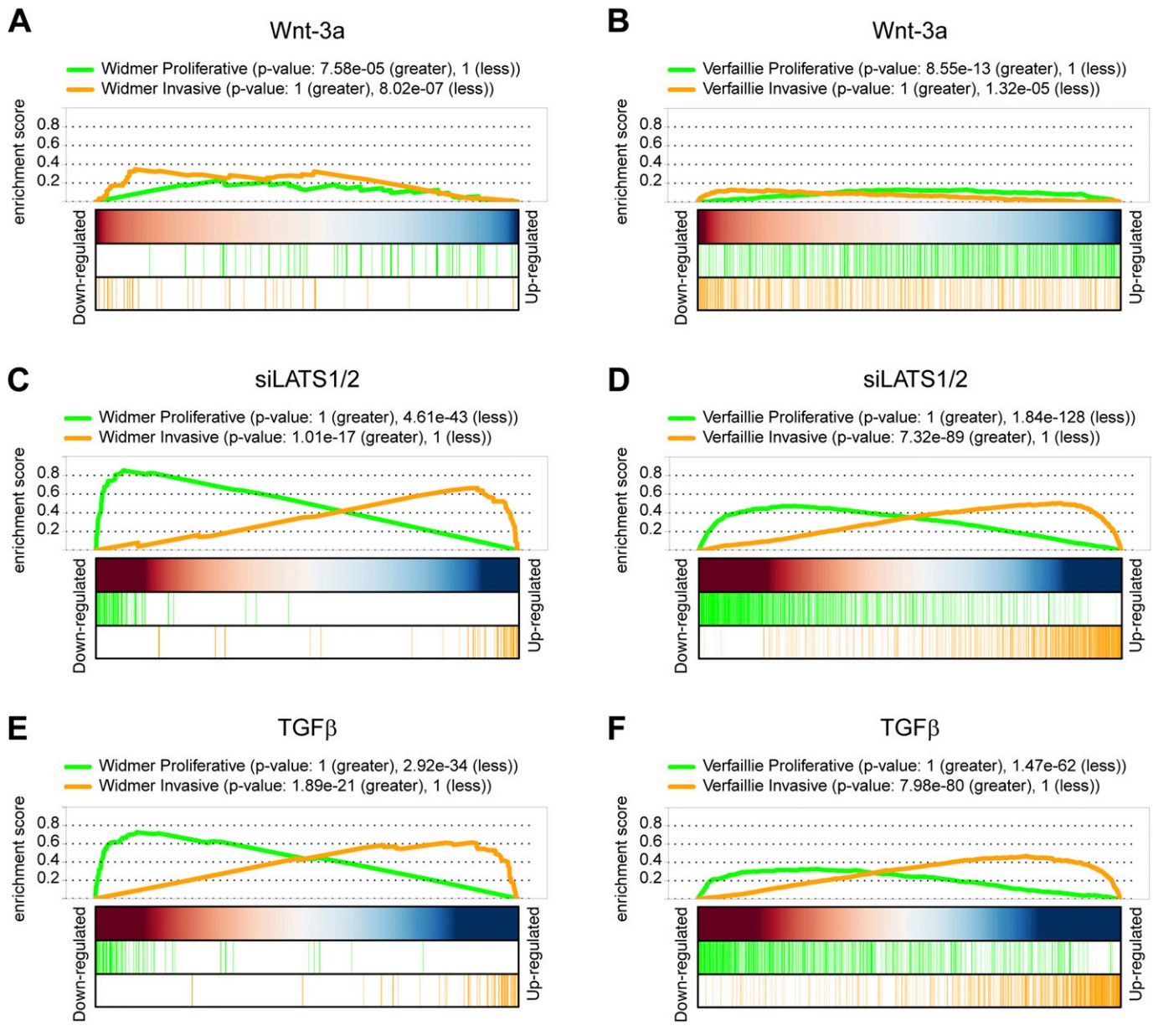

**Figure 5. Visualization of the gene set enrichment of proliferative and invasive gene signatures.**
**(A, B)** Enrichment during Wnt-3a treatment. **(C, D)** Enrichment during siLATS1/2 treatment. **(E, F)** Enrichment during TGFβ treatment. **(A, B, C, D, E, F)** Gene set enrichment analysis was performed against the gene sets described by Widmer et al (2012) (A, C, E) and Verfaillie et al (2015) (B, D, F).

well as the global gene expression analyses (Figs 3–5). Hierarchical clustering analysis of the genes downstream of Wnt-3a, that is, the genes that respond in their expression to TGFβ, siLATS1/2, and Wnt-3a with a posterior probability of >90%, revealed distinct clusters of genes (Fig 6C and Table S1). Consistent with our previous analyses, a large number of these commonly regulated genes were found to be regulated in the same direction by all treatments. Another large number of genes were regulated in one direction by Wnt-3a and in the opposite direction by TGFβ and siLATS1/2, consistent with the results presented above, thus confirming the robustness of the NEM. Hence, NEM analysis established a clear hierarchy between SMAD, YAP/TAZ, and β-catenin transcriptional control in melanoma phenotype switching.

## Discussion

The crosstalk between various signaling pathways and their transcriptional effectors is a common theme underlying cell plasticity in cell lineage transitions in development and disease, including phenotype switches and cancer cell EMT. These interactions are complex and the hierarchies between individual pathways are difficult to decipher. Using in vitro perturbation studies, global gene expression analysis, and NEM computation, we revealed the hierarchy of three major signaling pathways converging to each other during melanoma phenotype switching.

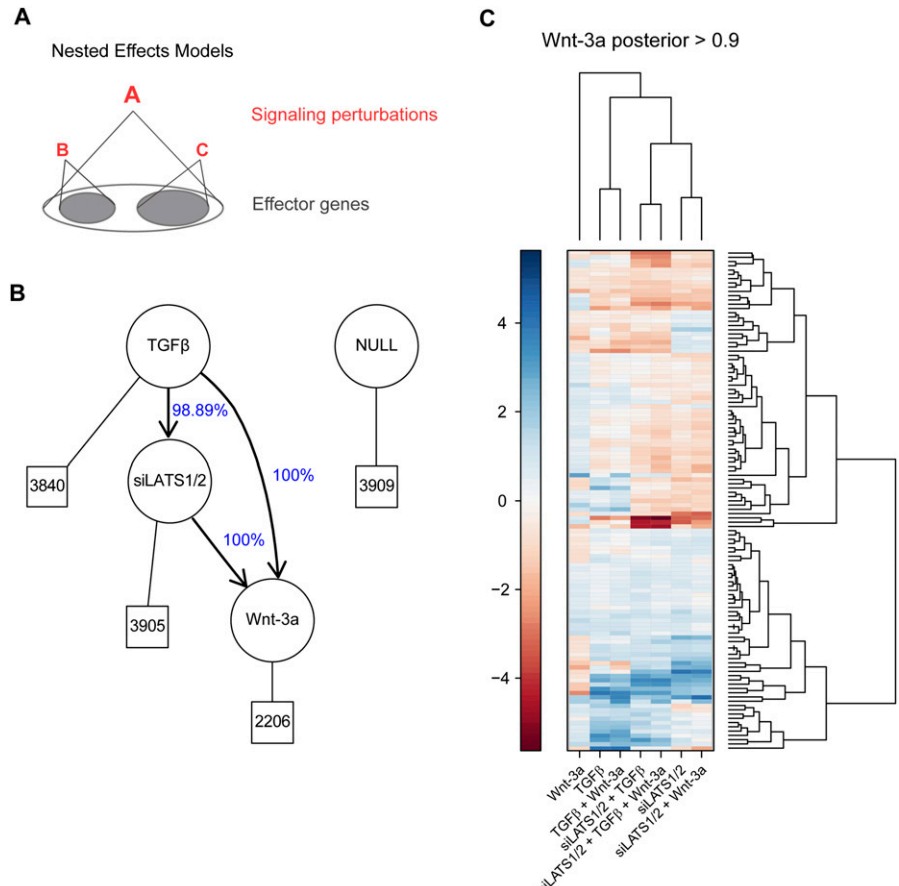

**Figure 6. Epistasis of TGFβ/SMAD, YAP/TAZ, and Wnt/β-catenin signaling as computed by nested effects models (NEMs).**
**(A)** Schematic diagram of an NEM. Factor A is predicted upstream of both factors B and C if perturbation of A affected transcripts (effectors) that were also affected by perturbation of factors B and C. **(B)** Optimal NEM of the three treatments TGFβ, siLATS1/2, and Wnt-3a based on single and combinatorial treatments. Edges are labelled for their bootstrap support. For example, the edge from TGFβ to siLATS1/2 appeared in 9,889 of 10,000 runs. Hence, in 111 runs, TGFβ and siLATS1/2 were not directly connected, but parallel to each other. Boxes indicate the number of genes reacting to the specific NEM regulator: TGFβ: genes reacting to TGFβ only; siLATS1/2: genes reacting to TGFβ and siLATS1/2; Wnt-3a: genes reacting to all three single treatments. **(C)** Hierarchical clustering of the genes corresponding to the Wnt-3a regulator (genes regulated by TGFβ, siLATS1/2, and Wnt-3a) with a posterior probability of greater than 0.9. Heat map shows log₂ fold changes.

We found that TGFβ signaling as well as activation of YAP/TAZ induced by knockdown of LATS1/2 led to strong induction of a proliferative-to-invasive phenotype switch, whereas canonical Wnt/β-catenin signaling rather promoted the proliferative phenotype of patient-derived melanoma cells, yet was required but not sufficient for the phenotype switch downstream of TGFβ and YAP/TAZ signaling. TGFβ activates a variety of signal transduction cascades, most notably canonical SMAD2/3 signaling, and is one of the most potent inducers of transcriptional reprogramming. We found TGFβ/SMAD signaling to act upstream of YAP/TAZ and β-catenin, thereby confirming the importance of TGFβ signaling in melanoma phenotype switching. However, activation of YAP/TAZ by means of LATS1/2 knockdown also resulted in a substantial change in gene expression which was independent of SMAD activities, yet it was also observed upon TGFβ treatment, consistent with the notion that YAP/TAZ signaling exerted a subprogram downstream of TGFβ signaling. Notably, in the human melanoma cells used, TAZ, and to a lesser extent YAP, was required for both siLATS1/2 and TGFβ-induced proliferative-to-invasive phenotype switching. Together, these data suggest that YAP/TAZ are critical inducers of the invasive phenotype and central regulators of a subprogram of canonical TGFβ/SMAD–induced phenotype switching.

The results from gain- and loss-of-function experiments also indicated that canonical Wnt/β-catenin signaling promoted a proliferative, differentiated phenotype, which supported the association of β-catenin with a favorable clinical prognosis. This observation is consistent with previous reports and might be a direct consequence of β-catenin inducing the expression of MITF, the key transcriptional regulator of melanocyte differentiation and of a proliferative phenotype in melanoma cells (Damsky et al, 2011; Widmer et al, 2012; Hartman & Czyz, 2015). On the other hand, we found that β-catenin was still required for the proliferative-to-invasive phenotype switch, suggesting that Wnt/β-catenin exerted a dual function in melanoma cells. Differential gene expression analysis revealed that a large number of genes was differentially regulated by TGFβ treatment and knockdown of LATS1/2, yet also by Wnt signaling. Most these genes followed two prominent gene expression patterns, either regulated in the same direction by all treatments or regulated in one direction by both TGFβ and siLATS1/2 but in the opposite direction by Wnt-3a treatment. As a transcriptional activator, β-catenin has been found in complexes together with SMAD2/3 and YAP/TAZ (Varelas et al, 2010; Azzolin et al, 2012; Attisano & Wrana, 2013; Piersma et al, 2015a, 2015b). This suggests that β-catenin might contribute to the expression of a subset of genes required for a proliferative-to-invasive phenotype switch. Consistent with this, functional experimentation and NEM analysis positioned β-catenin downstream of TGFβ/SMAD and YAP/TAZ signaling. Moreover, systematic combinations of the treatments showed that TGFβ treatment and LATS1/2 knockdown are dominant over canonical Wnt/β-catenin signaling and strongly counter-regulate a variety of Wnt-3a target genes, particularly including

phenotype-specific markers. Together, these observations suggest that $\beta$-catenin may play context-dependent roles and that it contributes to melanoma progression in the context of active SMAD and YAP/TAZ signaling, which may be distinct from its original function in canonical Wnt signaling. This conclusion is also consistent with the experimental observation that the loss of N-cadherin expression has not substantially affected gene expression and melanoma phenotype switching. These observations are comparable to previous reports by our laboratory on the contribution of TGF$\beta$, YAP/TAZ, and canonical Wnt signaling to EMT in breast cancer cells, where TGF$\beta$ has been found to be epistatic to YAP/TAZ, and where Wnt/$\beta$-catenin signaling also exerts a critical yet rather auxiliary function to both cell proliferation and EMT (Diepenbruck et al, 2014; Meyer-Schaller et al, 2019; Buechel et al, 2021).

There is extensive crosstalk and feedback not only between the three pathways examined here, but likely by many more. In particular, TGF$\beta$ and Wnt ligands may activate a variety of non-canonical pathways. At which levels TGF$\beta$/SMAD, Hippo/YAP/TAZ, and Wnt/$\beta$-catenin impinge on each other and whether these interactions are direct or indirect remains to be investigated. For example, these transcription factors could be part of the same transcriptional complexes binding to gene promoters, they could interact with each other by binding to different DNA motifs in the context of super-enhancers, or their transcriptional activities could be indirectly regulated by the other signaling pathways by post-translational modifications affecting protein stability, signal transduction, or transcriptional output. Of course, the interactions between different signaling pathways in the regulation of gene expression may differ from gene to gene, between the phenotypic states of cells, and between cell types. Thus, future studies will be required to delineate the exact mechanisms underlying this crosstalk.

Here, we have demonstrated that SMAD, YAP/TAZ, and $\beta$-catenin are required for a proliferative-to-invasive phenotype switch in melanoma cells. We have subsequently dissected the global hierarchy between these signaling pathways and we have found that TGF$\beta$/SMAD signaling is epistatic to YAP/TAZ signaling but that the transcriptional activities of both strongly induce proliferative-to-invasive phenotype switching in melanoma cells. Hence, these pathways may offer suitable therapeutic targets to interfere with melanoma progression and ultimately prevent metastasis formation and drug resistance. Here, YAP/TAZ may be of special interest as they are critical regulators that may be activated not only by TGF$\beta$ signaling but potentially by a variety of other stimuli. In contrast, even though $\beta$-catenin appears required for phenotype switching, it exerts context-dependent dual functions and thus may be a less suitable target.

# Materials and Methods

## Cell culture

M000921 and M010817 patient-derived melanoma cells (Hoek et al, 2006) were obtained from R Dummer (University Hospital Zürich). Cells were cultured in RPMI1640 (Sigma-Aldrich) supplemented with 10% heat-inactivated fetal bovine serum (Sigma-Aldrich), 5 mM L-glutamine (Sigma-Aldrich), 1 mM sodium pyruvate (Thermo Fisher Scientific), 100 U penicillin, and 0.1 mg/ml streptomycin (Sigma-Aldrich). Cells were kept in a humidified incubator at 37°C and 5% $CO_2$. Micrographs were acquired with a Leica DMIL phase-contrast microscope with 10×, 0.22 NA objective and were processed with ImageJ (ImageJ, Wayne Rasband, National Institutes of Health).

## siRNA-mediated knockdown and growth factor treatments

Cells were reverse transfected with non-targeting control or gene-specific siRNA pools (Table S2) using lipofectamine RNAiMAX (Invitrogen) according to the manufacturer's instructions. 1 d post transfection, cells were treated with 2 ng/ml of recombinant human TGF$\beta$1 (#240-B; R&D Systems), and/or 75 ng/ml recombinant murine Wnt-3a (#315-20; PeproTech). For gene expression analysis by quantitative RT-PCR or RNA sequencing, cells were treated for 2 d. For migration and 3D Matrigel invasion assays, cells were treated for a total of 5 d and were forward re-transfected with siRNAs after 3 d to sustain knockdown. The source of siRNAs is given in Table S2.

## Trans-well migration assay

M010817 cells were pretreated with siRNA targeting siLATS1/2 or non-targeting control and TGF$\beta$ or Wnt-3a for 5 d as described above. Cells were washed twice with PBS, trypsinized and resuspended in medium containing 0.1% FBS. $5 \times 10^4$ cells were centrifuged and resuspended in 300 $\mu$l medium containing 0.1% FBS and seeded into Falcon migration transwell inserts with 8.0 $\mu$m PET membrane (353097; Corning) in technical duplicates. 700 $\mu$l medium containing 10% FBS was added to the bottom chamber as a chemoattractant. TGF$\beta$ or Wnt-3a were added to the top and bottom chamber for the respective samples. As a control, 300 $\mu$l of the same cell suspensions plus 700 $\mu$l medium containing 10% FBS were seeded into 24-wells in duplicates. After 20 h the cells were fixed with 4% paraformaldehyde for 15 min, washed three times with PBS and cell nuclei were stained with DAPI (1 $\mu$g/ml; Sigma-Aldrich). Non-migrated cells on the upper side of the membrane were removed with a cotton swab. Migrated cells on the bottom-side of the membrane as well as the control plate were imaged with a Leica DMI 4000 microscope with a 10×, 0.3 NA objective. 10–15 images of migrated cells were acquired and quantified per condition, and the number of migrated cells was normalized to control wells (five images per condition). Graphs display mean values of three biological replicates and error bars represent SEM. Two-tailed paired $t$ tests were performed.

## 3D Matrigel invasion assay

M010817 cells were pretreated with siRNA targeting siLATS1/2 or non-targeting control and TGF$\beta$ or Wnt-3a for 5 d as described above. $3.4 \times 10^4$ cells in 50 $\mu$l medium containing TGF$\beta$ or Wnt-3a where appropriate were seeded into angiogenesis $\mu$-Slide (Ibidi) coated with 10 $\mu$l growth factor-reduced Matrigel (diluted 1 + 2; BD Biosciences, 356230). Cells were incubated for 4 h and imaged using a Leica DMIL phase-contrast microscope with 10×, 0.22 NA objective. Images were processed with ImageJ (ImageJ, Wayne Rasband, National Institutes of Health).

## RNA isolation and quantitative RT-PCR

Total RNA was isolated using TRI reagent (Sigma-Aldrich) according to the manufacturer's instruction. cDNA was synthesized using ImProm-II Reverse Transcriptase (Promega) according to the manufacturer's instructions. Quantitative PCR was performed on a StepOnePlus Real-Time PCR System (Applied Biosystems) using Power-Up SYBR green (Applied Biosystems). Assays were performed in duplicates and target gene expression levels were normalized to *RPL19*. Fold changes were calculated using the comparative Ct method ($\Delta\Delta$Ct). Data were analyzed using GraphPad Prism 7.0 (GraphPad Software, Inc.). Graphs display mean values of three biological replicates and error bars represent SEM. Two-tailed ratio-paired *t* tests were performed. Primer sequences are listed in Table S3.

## Immunoblotting analysis

Cells were lysed in RIPA buffer (Merck) supplemented with 2 mM NaF, 2 mM orthovanadate, 1 mM DTT, and 1× protease inhibitor cocktail (Sigma-Aldrich) or by boiling in lysis buffer (300 mM Tris–HCl pH 6.8, 6% SDS, and 25% glycerol) for 5 min at 95°C. Lysates were sonicated and protein concentrations were measured using Bio-Rad Bradford solution (Bio-Rad Laboratories) according to the manufacturer's instructions. Equal amount of protein was resolved by SDS–PAGE (8% or 10% gels) and transferred to nitrocellulose membrane (Amersham Protran; Sigma-Aldrich) or polyvinylidene difluoride membrane (Carl Roth GmbH) by wet-transfer. Membranes were blocked in 5% milk or 5% BSA diluted in 0.05% Tween-20 in PBS (PBS-T) for 1 h at room temperature followed by incubation with primary antibodies (Table S4) diluted in blocking solution overnight at 4°C and incubation with horseradish peroxidase conjugated secondary antibodies (Jackson ImmunoResearch Labs) diluted 1:10,000 in 5% milk in PBS-T for 1 h at room temperature. Protein was detected by chemiluminescence using a Fusion F67 chemiluminescence reader (Vilber Lourmat). The source of antibodies used is given in Table S4.

## Co-immunoprecipitation

Cells were treated with siRNA targeting siLATS1/2 or non-targeting control and TGFβ as described above. For immunoprecipitation, cells were washed with ice cold PBS, scraped off in 300 µl cold IP lysis Buffer (20 mM Tris pH 7.5, 150 mM NaCl, 10% glycerol, 1% NP-40, 2 mM EDTA, 2 mM NaF, 1 mM orthovanadate, protease inhibitor cocktail [Sigma-Aldrich], and 1 mM DTT) and transferred into a pre-chilled Eppendorf tube. Cells were lysed for 20 min rotating at 4°C. Samples were centrifuged for 10 min at 16,000*g* at 4°C and the supernatants were transferred to new tubes. Protein concentrations were measured using Bio-Rad Bradford solution (Bio-Rad Laboratories) according to the manufacturer's instructions. An input sample was collected. Per IP, 5 µg mouse-anti-β-catenin (clone 14/β-Catenin; BD Biosciences) or control IgG were diluted in 200 µl 0.02% Tween-20 (PBS-T). Per sample, 40 µl Dynabeads Protein G for Immunoprecipitation (Thermo Fisher Scientific) were washed twice with 500 µl PBS-T. Dynabeads were collected using a magnetic stand and the antibody dilution was added followed by 15 min of rotation at room temperature. Antibody-bound beads were collected using a magnetic stand, the supernatant was aspirated, and 0.5 mg of protein lysate was added and incubated overnight at 4°C while rotating. The next day, the beads were washed three times with 500 µl IP wash buffer (15 mM Tris–HCl, pH 7.8, 100 mM NaCl, 2 mM NaF, 1 mM orthovanadate, protease inhibitor

cocktail [Sigma-Aldrich], and 1 mM DTT) and eluted in 40 µl 1× SDS–PAGE sample buffer (10% glycerol, 2% SDS, 65 mM Tris, 0.01 mg/ml bromophenol blue, and 1% β-mercaptoethanol) by boiling at 95°C for 5 min. Empty beads were removed and the supernatant and the input sample were analyzed by immunoblotting.

## Immunofluorescence staining of cultured cells

Cells grown on uncovered glass coverslips (#1, round, 10 mm; Thermo Fisher Scientific) were washed with PBS and fixed with 4% paraformaldehyde for 20 min at room temperature, followed by permeabilization with 0.5% NP-40 for 5 min and blocking with 3% BSA for 1 h. Cells were incubated with primary antibodies (mouse-anti-β-catenin clone 14/β-Catenin; BD Biosciences and rabbit–anti–N-cadherin H-63; Santa Cruz Biotechnology) diluted 1:100 in 3% BSA overnight at 4°C followed by incubation with a fluorophore-coupled secondary antibody (Alexa Fluor 488 or 568, 1:400; Invitrogen) for 1 h at room temperature in the dark. Cell nuclei were stained with DAPI (1 µg/ml; Sigma-Aldrich), and coverslips were mounted with DAKO fluorescence mounting medium (Agilent). Images were acquired with a Leica DMI 4000 microscope with 40×, 1.3 NA objective and were processed with ImageJ.

## RNA sequencing analysis

M000921 and M010817 cells were reverse-transfected with siRNA targeting LATS1 and LATS2 or non-targeting control. Transfected cells were treated with TGFβ and/or Wnt-3a for 2 d starting 1 d post transfection. Samples were prepared as technical duplicates and total RNA was isolated using the miRNeasy Mini Kit (QIAGEN) with on-column DNA digestion using RNase-Free DNase Set (QIAGEN) according to the manufacturer's instructions.

Library preparation was done using Truseq Stranded mRNA sample prep from Illumina with 200 ng RNA as input. Sequencing was carried out on Novaseq with Single Read conditions (Read 1: 101/Index 1 : 8/Index 2: 8). Single-end RNA-seq reads (81-mers) were obtained and mapped to the human genome assembly, version hg19 (GRCh37.75), with RNA-STAR (Dobin et al, 2013), with default parameters except for allowing only unique hits to the genome (outFilterMultimapNmax = 1) and filtering reads without evidence in spliced junction table (outFilterType = "BySJout"). Expression levels per gene (counts over exons) for the RefSeq mRNA coordinates from UCSC (genome.ucsc.edu, downloaded in December 2015) were quantified using qCount function from the Bioconductor R package QuasR (version 1.12.0) (Gaidatzis et al, 2015).

## Preprocessing and differential gene expression analysis

We used the following linear model for the expression *y* of a gene across experiments,

$$y = \beta_0 + \beta_1 L + \beta_2' TGF\beta' + \beta_3' siLATS1\big/2' + \beta_4' Wnt3a'$$
$$+ \beta_5' TGF\beta + siLATS1\big/2' + \beta_6' TGF\beta + Wnt3a'$$
$$+ \beta_7' siLATS1\big/2 + Wnt3a' + \beta_8' TGF\beta + siLATS1\big/2 + Wnt3a'$$

with covariate *L* for the second cell line and the double and triple perturbations as independent variables, that is, we performed

pairwise comparisons accounting for the difference in cell lines. For example, the gene expression is $\beta_0$ in the control experiments of the first cell line and $\beta_0 + \beta_1$ in the second cell line. In the single TGF$\beta$ perturbation in the first cell line, the gene expression is $\beta_0 + \beta_2$, whereas it is $\beta_0 + \beta_5$ in the double perturbation of TGF$\beta$ and siLATS1/2. Hence, the differential expression in the single TGF$\beta$ perturbation is $\beta_2$ accounting for the cell line effect.

For the visualization in the principle component analysis we used the batch removal function of the Bioconductor R package "limma" (Ritchie et al, 2015) to remove the cell line effect $L$ from the raw data.

We employed the same linear model for the differential expression analysis with the Bioconductor R package "edgeR" (Robinson et al, 2010). P-values were corrected by Benjamini and Hochberg (1995). We binarized the corrected P-values by a cutoff of 10%.

The Venn diagram was drawn with the CRAN R package "venn" (https://cran.r-project.org/web/packages/venn/index.html).

NEMs (Markowetz et al, 2007) use log densities for each E-gene, S-gene pair as data input to compute the hierarchical network. A large value (>0) denotes that the E-gene is affected by the knockdown of the S-gene, and a low value (<0) denotes the E-gene was not affected. For a given network, NEM computes a predicted profile of effects (1) and no effects (0) for each E-gene, S-gene pair, and scores it against the log densities, for example, a network scores highly, if many negative log densities match predicted 0s and positive log densities match predicted 1s. We fitted a $\beta$-uniform mixture model with the optimizing function from the Bioconductor R package "BioNet" (Beisser et al, 2010) to the P-values derived from edgeR and computed the log densities using the density function of the $\beta$ component (Fig S18).

We applied NEM implemented in the Bioconductor R package "mnem" (Pirkl & Beerenwinkel, 2018) to exhaustively compute the optimal network of the three S-genes TGF$\beta$, siLATS1/2 and Wnt-3a, and the bootstrap support of all edges.

### Top gene expression patterns

We counted genes following each of the 128 possible differential expression patterns over the seven treatments based on fold change direction. The heat maps in Fig 4 are based on the following ranking: we filtered for genes with a corrected P-value of less than 10%. We ranked the genes after their absolute log fold change in the Wnt-3a single treatment. Next, we filtered for the specific gene expression patterns 1 and 2.

### Gene set enrichment analysis

The P-values for rank-based enrichment were computed with the Wilcoxon rank sum test ("stats" package, R Core Team, 2020). The genes were ranked by their log fold change. The curve to visualize the enrichment follows the function:

$$f(i) = \left| \frac{rank(i)}{|n|} - \frac{i}{|m|} \right|,$$

with $i$ the rank of the ith gene among the $m$ genes of interest and $rank(i)$ the rank of the ith gene among all $n$ global genes. The enrichment analysis and visualization is implemented in the

Bioconductor R package "epiNEM" (Pirkl et al, 2017). The gradient of the fold changes was cutoff at absolute 1 for improved visualization.

Entrez ids were converted to gene symbols with the Bioconductor R package "biomaRt" (Durinck et al, 2009). Entrez ids were kept when no matching gene symbol was recovered.

## Data Availability

Further information and requests for resources and reagents should be directed to and will be fulfilled by the Lead Contact, Gerhard Christofori (gerhard.christofori@unibas.ch).

The RNA sequencing data from this publication have been deposited to National Center for Biotechnology Information's (NCBI) Gene Expression Omnibus and assigned the identifier GSE188463 (https://www.ncbi.nlm.nih.gov/geo/query/acc.cgi?acc=GSE188463).

## Supplementary Information

## Acknowledgements

We thank R Dummer (University Hospital Zürich) for providing M000921 and M010817 cells, C Beisel and the Genomics Facility Basel for RNA sequencing, and P Lorentz and the Department of Biomedicine (DBM) Microscopy Facility for support with microscopy. This work was supported by a Swiss National Science Foundation Sinergia Grant (CRSII3_154412CRSII3_154412) and SystemsX.ch RTD MERiC.

### Author Contributions

F Lüönd: conceptualization, resources, data curation, formal analysis, validation, investigation, visualization, methodology, and writing—original draft, review, and editing.
M Pirkl: conceptualization, resources, data curation, software, formal analysis, validation, investigation, visualization, methodology, and writing—original draft.
M Hisano: data curation, formal analysis, validation, investigation, methodology, and writing—review and editing.
V Prestigiacomo: conceptualization, data curation, formal analysis, investigation, methodology, and writing—original draft.
RKR Kalathur: resources, data curation, software, formal analysis, validation, methodology, and writing—original draft.
N Beerenwinkel: conceptualization, resources, software, formal analysis, supervision, funding acquisition, investigation, and writing—original draft.
G Christofori: conceptualization, resources, data curation, formal analysis, supervision, funding acquisition, validation, investigation, visualization, project administration, and writing—original draft, review, and editing.

## Conflict of Interest Statement

The authors declare that they have no conflict of interest.

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
