## [Reviewer comments · Life Science Alliance]

Life Science Alliance

Hierarchy of TGF β /SMAD, Hippo/YAP/TAZ and Wnt/ β -catenin signaling in melanoma phenotype switching

Fabiana Lüönd, Martin Pirkl, Mizue Hisano, Vincenzo Prestigiacomo, Ravi Kalathur, Niko Beerenwinkel, and Gerhard Christofori
DOI: <https://doi.org/10.26508/lsa.202101010>

Corresponding author(s): Gerhard Christofori, University of Basel

Review Timeline:

Submission Date:	2021-01-05
Editorial Decision:	2021-03-22
Revision Received:	2021-10-19
Editorial Decision:	2021-11-05
Revision Received:	2021-11-11
Accepted:	2021-11-12

Transaction Report:

March 22, 2021

Re: Life Science Alliance manuscript #LSA-2021-01010-T

Prof. Gerhard Christofori
University of Basel
Dept. of Biomedicine
Mattenstrasse 28
Mattenstrasse 28
Basel CH-4058
Switzerland

Dear Dr. Christofori,

Thank you for submitting your manuscript entitled "Hierarchy of TGF /SMAD, Hippo/YAP/TAZ and Wnt/ -catenin signaling in melanoma phenotype switching" to Life Science Alliance. We apologize for this extended and unusual delay in getting back to you. The manuscript was assessed by expert reviewers, whose comments are appended to this letter.

As you will note from the reviewers' comments below, the reviewers are overall enthusiastic about the findings, but have pointed out some concerns and requested some additional data that should be addressed prior to further consideration of this manuscript at LSA. In particular, both reviewers 2 and 3 point out that the manuscript would be drastically improved by additional analysis of biological relevance of these findings - these could be done in accordance to the experiments suggested by Reviewer 3. Along with this, all 3 reviewers have asked for some additional controls and other minor edits that should also be addressed. We would, thus, like to invite you to submit a revised version of the manuscript that addresses the reviewers' concerns.

To upload the revised version of your manuscript, please log in to your account: <https://lsa.msubmit.net/cgi-bin/main.plex>
You will be guided to complete the submission of your revised manuscript and to fill in all necessary information. Please get in touch in case you do not know or remember your login name.

Thank you for this interesting contribution to Life Science Alliance. We are looking forward to receiving your revised manuscript.

Sincerely,

Shachi Bhatt, Ph.D.
Executive Editor
Life Science Alliance
<https://www.lsa-journal.org/>
Tweet @SciBhatt @LSAjournal

B. MANUSCRIPT ORGANIZATION AND FORMATTING:

Reviewer #1 (Comments to the Authors (Required)):

Lüönd et al. analyzed the proliferative-to-invasive phenotype switch that occurs during malignant melanoma progression. To this end, the authors used two patient-derived melanoma cell lines and tested for the potential of TGF β , Hippo and Wnt pathways to contribute to the phenotype switching using Tgfb and Wnt ligands as well as siLATS1/2 to activate each pathway individually or in combination. Analysis of specific target gene expression, representing activity of each signaling pathway combined with global transcriptome analysis and computational nested effects models demonstrated that all pathways contribute to the proliferative-to-invasive phenotype switch, but to different extents. Moreover, a hierarchical dependency of the pathways was extracted from the computational models with TGF β on top regulating the YAP/TAZ pathway and Wnt/b-catenin situated further downstream in the cascade. Interestingly, the Wnt/b-catenin pathway also displayed a dual role, regulating expression of proliferative genes, but also enhanced the expression of invasive specific genes during the phenotypic switch.

Understanding of melanoma progression and how different pathways converge during malignant progression is very crucial and our knowledge about how signaling pathways crosstalk to each other and how they are hierarchically organized is still limited. Lüönd and colleagues elegantly addressed this question and provided evidence how Tgfbeta, YAP/TAZ and Wnt signals converge and how they are hierarchically organized during the phenotype switch. The analysis is done very accurately, using very elegant experimental systems and controls to prove the hypothesis and dissect the interplay of the different pathways. The manuscript is well-written with clearly defined hypotheses and appropriate conclusions extracted from the data. This important analysis will be interesting for a large community of scientists in the melanoma, tumor biology and EMT research area and merits publication in Life Science Alliance. I have only a few comments to be addressed before publication:

1. In Figs. 2a, S3a and following the authors make use of cells that harbor siRNA knockdowns of TAZ, YAP, SMAD4 and CTNNB1. The authors claim that "knockdown of SMAD4, TAZ and b-catenin, but not of YAP, significantly counteracted the TGF β -induced" effects and loss of TAZ and b-catenin "significantly counteracted the siLATS1/2-induced phenotype switch". Especially for the TGF β experiment it is important to also show the effect of siTAZ, siYAP, siSMAD4 and siCTNNB1 alone in unstimulated conditions to draw these conclusions. Does the knockdown alone already affect target gene expression?
2. Suppl. Fig. 4. Although the authors use this experiment to exclude a contribution of cadherin-bound b-catenin to the observed effects, it would be interesting to see whether siCDH2 only results in reduction of cadherin-bound b-catenin or whether also Wnt/b-catenin signaling is affected, i.e. reduced by inefficient total b-catenin amounts.
3. Are the observations very specific to the proliferative-to-invasive phenotype switch or can they be applied to any other form of tumor cell EMT? A brief discussion of this aspect would help to put the data into context.
4. In the pdf and word doc version, most of the equations in the Materials and Methods part are not displayed properly

Reviewer #2 (Comments to the Authors (Required)):

The authors have studied the relative contribution of the YAP/TAZ, Wnt/b-catenin and TGF β /SMAD signaling in the switching of melanocytes from the proliferative to the invasive phenotype. Their experimental approach was based in the analysis of RNA expression in cells with specific depletion of essential genes in these pathways and the figures correspond to the analysis of individual genes or the in silico analysis of RNA-Seq. The most relevant conclusion suggests that although b-catenin transcriptional activity is not sufficient it is required for this switch. I consider that the article is a bit limited and needs more experimental work to respond to some concerns and to better validate the conclusions with additional controls. I indicated below some issues that need to be addressed before publication:

- There is a general lack of interest in determining the physiological relevance of their findings. Their conclusions should be validated with experiments assessing the invasive or metastatic capability of the cells with lack or gain of function of the studied signaling pathway, for instance Wnt/b-catenin.
- The poor effect of Wnt3a on the activation of mesenchymal genes and the requirement of b-catenin for the expression of TGF- β or TAZ-dependent genes suggest that the activation of these genes is not dependent of the canonical b-catenin-associated transcriptional factors Tcf4 or LEF1. This point is relevant and should be studied determining the effect of the depletion of these genes or using a dominant-negative form of Tcf4 or Lef1.
- The activation of Wnt-dependent genes by Wnt3a is lower than by TGF β . This is surprising and suggests that the Wnt3a pathway is constitutively activated in these cells. The authors should check this, at least analyzing b-catenin stabilization and translocation to the nucleus in the absence or presence of Wnt3a.
- The results presented in Suppl Figs 4 and 5 do not provide any relevant information for the article and should be deleted, unless additional experiments are performed demonstrating that indeed b-catenin is interacting with N-cadherin. The title of this figures is that "b-catenin-mediated adhesion is not required for phenotype switching" whereas the results show that N-cadherin is not required for this switch. Moreover, these experiments present the problem that they have been performed with a not-specified siRNA that is not available anymore; so, they cannot be reproduced.
- The efficiency in the siRNA-mediated depletion of the different genes should be assessed by Western blot, that is more indicative than RT-PCR.
- The authors need to provide a much better description of the cell lines used in these assays and in general, of the conditions used. Have the cell lines been authenticated?
- How confident are the authors that their siRNA do not present off-target effects? Only a combination of siRNAs has been used to deplete each target. At least some results (those obtained with b-catenin-depletion) should be rescued with a siRNA-resistant ectopic gene or validated using an alternative shRNA.
- Other technical issues. What is the source of Wnt3a? Is it recombinant or coming from a conditioned-medium? Another b-catenin-target gene with a better stimulation by Wnt3a than NKD1 should be included in the Figure 1b analysis since this gene is not significantly upregulated by Wnt3a.

Reviewer #3 (Comments to the Authors (Required)):

In the present ms. the authors analyze the hierarchical interaction of three signaling pathways, TGF β /SMAD, Hippo/YAP/TAZ, and Wnt/b-catenin, in the phenotypic switch of melanoma cells. By performing functional assays in two patient derived melanoma cells, combined with HTP sequencing and computational analyses (i.e, Nested Effect Models, NEM) they infer the hierarchical interaction among the mentioned pathways in the transition from the proliferative to the invasive phenotype. The main conclusion of the study is that transcriptional regulation by the three signaling pathways is required for the proliferative-to-invasive switch in melanoma. While TGF β /SMAD is on the top of the hierarchy followed by the Hippo pathway, and then by Wnt/b-catenin, b-catenin seems to play a dual role supporting both the proliferative phenotype and the invasive switch mediated by the other two pathways. This is an interesting study that provides novel information of basic mechanisms underlying the melanoma phenotypic switch and it can be of potential interest in the future for targeting specific pathways. The study is performed using state of the art experimental design and computational analyses in two melanoma cell lines. The results obtained after manipulating the signaling pathways in different combinations support the main conclusions of the authors on the hierarchical interaction of the pathways at transcriptional level. However, the main limitation of the study is the absence of biological studies to support that the transcriptional data indeed reflect distinct phenotypic states. At least some functional analyses of the effect of manipulation of the pathways on the phenotype should be included to increase the biological relevance of the present study. Other minor controls should be also considered before publication, as described below.

Main points:

1. The "invasive" phenotype mediated by TGF β treatment, and to a lower extent by Hippo activation, is only determined at morphological level described as a more mesenchymal phenotype (F1g. 1a; Suppl Fig S2a) and EMT marker analyses by RT-qPCR (Fig. 1b; Suppl Fig. S2b). At least some functional invasion assay, i.e., in vitro assays on matrigel Boyden chambers, needs to be included to fully support the claimed phenotypic switch. The same applies to cells treated with Wnt3A in which the claim of "not effect in cell morphology" is not easily appreciated only from the images provided (F1g. 1a; Suppl Fig S2a).
2. The strategy of using siControl melanoma cells to activate TGF β , Wnt and the Hippo pathways is correct to discard non-specific effects when using specific siRNAs (i.e, siLATS1/2). However, it should be important to show that similar phenotypic and transcriptomic effects are observed when parental melanoma cells are treated with the activating agents.

3. The RT-qPCR analysis of EMT-TFs expression in the different experimental situations should be extended to SLUG (SNAIL2), ZEB2 and TWIST1, at least after modulation of the individual pathways (Fig. 1b; Suppl Fig. S2b). As the expression of SNAIL2 and ZEB2 has been previously shown to be downregulated, in favor of ZEB1 and TWIST1, during the invasive switch in melanoma tumors by others (Caramel et al., *Cancer Cell*, 2013; Denecker et al., *Cell Death Differ*, 2014), the authors should check if similar changes apply or not to their cellular models.

Point-by-point reply to the reviewers' comments**Journal: Life Science Alliances****Manuscript: #LSA-2021-01010-T****The hierarchy of TGF β /SMAD, Hippo/YAP/TAZ and Wnt/ β -catenin signaling in melanoma phenotype switching**

Fabiana Lüönd, Martin Pirkl, Mizue Hisano, Vincenzo Prestigiacomo, Ravi K.R. Kalathur, Niko Beerenwinkel, and Gerhard Christofori

Introductory Remarks

We highly appreciate the constructive comments and suggestions by the editor and by the reviewers on our manuscript.

We have now spent the past 5 months to adequately address all the criticisms raised by the reviewers by a large number of additional experiments and by revising the manuscript and the figures accordingly, as detailed in the point-by-point reply.

In brief, we have performed experiments to assess the biological relevance of our findings by functional migration and invasion assays and to further clarify the role of β -catenin in melanoma phenotype switching. Moreover, we have performed additional control experiments to strengthen our conclusions and we have extended the description of the assays performed as well as the discussion of our findings.

As a consequence of these revisions, the Figures and Suppl. Figures have been revised and updated in their panels and the number of Appendix Figures has increased from 12 to 18 and from 3 to 4 Suppl. Tables. In addition, the presentation of the results has been revised in the text to accommodate the new data and to appropriately adapt the conclusions.

We apologize for the delay in our response, the current COVID crisis had a major and sometimes deleterious impact on the timely delivery of reagents and on the manpower and working conditions to execute the revisions.

The details of the revisions can be seen in the point-by-point reply to the reviewers' comments. We copied the reviewers' comments in *italic* and presented our reply in regular font.

Point-by-point reply**Reviewer #1**

Lüönd et al. analyzed the proliferative-to-invasive phenotype switch that occurs during malignant melanoma progression. To this end, the authors used two patient-derived melanoma cell lines and tested for the potential of TGF β , Hippo and Wnt pathways to contribute to the phenotype switching using Tgfb and Wnt ligands as well as siLATS1/2 to activate each pathway individually or in combination. Analysis of specific target gene expression, representing activity of each signaling pathway combined with global transcriptome analysis and computational nested effects models demonstrated that all

pathways contribute to the proliferative-to-invasive phenotype switch, but to different extents. Moreover, a hierarchical dependency of the pathways was extracted from the computational models with TGFbeta on top regulating the YAP/TAZ pathway and Wnt/b-cat situated further downstream in the cascade. Interestingly, the Wnt/b-cat pathway also displayed a dual role, regulating expression of proliferative genes, but also enhanced the expression of invasive specific genes during the phenotypic switch.

Understanding of melanoma progression and how different pathways converge during malignant progression is very crucial and our knowledge about how signaling pathways crosstalk to each other and how they are hierarchically organized is still limited. Lüönd and colleagues elegantly addressed this question and provided evidence how Tgfbeta, YAP/TAZ and Wnt signals converge and how they are hierarchically organized during the phenotype switch. The analysis is done very accurately, using very elegant experimental systems and controls to prove the hypothesis and dissect the interplay of the different pathways. The manuscript is well-written with clearly defined hypotheses and appropriate conclusions extracted from the data. This important analysis will be interesting for a large community of scientists in the melanoma, tumor biology and EMT research area and merits publication in Life Science Alliance. I have only a few comments to be addressed before publication:

We thank the reviewer for her/his interest in the study and for the constructive comments to improve the quality of the manuscript.

1. In Figs. 2a, S3a and following the authors make use of cells that harbor siRNA knockdowns of TAZ, YAP, SMAD4 and CTNNB1. The authors claim that "knockdown of SMAD4, TAZ and b-catenin, but not of YAP, significantly counteracted the TGFb-induced" effects and loss of TAZ and b-cat "significantly counteracted the siLATS1/2-induced phenotype switch". Especially for the TGFbeta experiment it is important to also show the effect of siTAZ, siYAP, siSMAD4 and siCTNNB1 alone in unstimulated conditions to draw these conclusions. Does the knockdown alone already affect target gene expression?

We appreciate the reviewer's question. Since we have observed that YAP/TAZ and β -catenin are required for the phenotype switch induced by loss of LATS1/2 (i.e. activation of YAP/TAZ) or TGF β (i.e. activation of SMAD4), we did not include the experimental results in ablating YAP/TAZ, SMAD4 or β -catenin in the absence of TGF β . As expected, siRNA-mediated depletion of YAP, TAZ, SMAD4e or β -catenin, had no significant effect on the expression of most melanocytic differentiation genes or mesenchymal marker genes. Only the melanocytic differentiation marker genes MITF and MLANA were found to be expressed at significantly higher levels in the absence of TAZ and β -catenin, confirming that these two factors promote the phenotype switch and their absence prevent it. However, none of the depletions resulted in a full phenotype switch as observed with TGF β or the ablation of LATS1/2. These results are now shown as **new Suppl. Fig. 4a,b and in Suppl. Fig. 6a,b**.

2. Suppl. Fig. 4. Although the authors use this experiment to exclude a contribution of cadherin-bound b-cat to the observed effects, it would be interesting to see whether siCDH2 only results in reduction of cadherin-bound b-cat or whether also Wnt/b-cat signaling is affected, i.e. reduced by inefficient total b-cat amounts.

To address this question, we have now assessed the expression of Wnt target genes upon activation of YAP/TAZ by siLATS1/2 and upon TGF β treatment with and without siRNA-mediated depletion of CDH2 (N-cadherin). We reasoned that the loss of N-cadherin releases β -catenin from the adherens junctions and thus may increase canonical WNT signaling. As is now shown in the **new Suppl. Fig. 8a,b, Suppl. Fig. 9b,c and Suppl. Fig. 10a,b**, there has not been any significant effect on the expression of mesenchymal marker genes and on Wnt target genes by the loss of N-cadherin expression, suggesting that the amount of β -catenin is not limited by N-cadherin-mediated adhesion in the mesenchymal-like cells. Rather, melanocytic marker genes seemed to be upregulated upon depletion of N-cadherin, further supporting the notion that β -catenin supports the proliferative state.

3. Are the observations very specific to the proliferative-to-invasive phenotype switch or can they be applied to any other form of tumor cell EMT? A brief discussion of this aspect would help to put the data into context.

Previous work in our laboratory on the contribution of canonical Wnt signaling and TGF β and YAP/TAZ signaling to EMT in breast cancer cells has shown that these pathways exert comparable activities as observed in the melanoma phenotype switch reported here. We have now included these results **into the Discussion section**, as proposed by the reviewer.

Diepenbruck et al., 2014, J. Cell Science 127, 1523-1536.

Meyer-Schaller et al., 2019, Dev. Cell 48, 539-553.

Buechel et al., 2021, PNAS 118, e 2020227118.

4. In the pdf and word doc version, most of the equations in the Materials and Methods part are not displayed properly.

We thank the reviewer for asking for this clarification. The presentation of the mathematical and computational analysis in the Methods section has now been revised accordingly. We hope that now the layout is stable during transmission.

Reviewer #2

The authors have studied the relative contribution of the YAP/TAZ, Wnt/ β -catenin and TGF β /SMAD signaling in the switching of melanocytes from the proliferative to the invasive phenotype. Their experimental approach was based in the analysis of RNA expression in cells with specific depletion of essential genes in these pathways and the figures correspond to the analysis of individual genes or the in silico analysis of RNA-Seq. The most relevant conclusion suggests that although β -catenin transcriptional activity is not sufficient it is required for this switch. I consider that the article is a bit limited and needs more experimental work to respond to some concerns and to better validate the conclusions with additional controls. I indicated below some issues that need to be addressed before publication:

We appreciate for the reviewer's comments and suggestions. We have now substantially

- There is a general lack of interest in determining the physiological relevance of their findings. Their conclusions should be validated with experiments assessing the invasive or metastatic capability of the cells with lack or gain of function of the studied signaling pathway, for instance Wnt/b-catenin.

We thank the reviewer for this suggestion. We have now performed a modified Boyden chamber migration assay which demonstrates that TGF β but neither siLATS1/2 nor WNT3a are able to promote cell migration (**new Fig. 1d**).

In addition, we have performed 3D invasion assays in Matrigel culture conditions. These assays have shown increased invasive growth by the mesenchymal melanoma cells as compared to the proliferative melanoma cells, as reported previously by our laboratory (Schlegel et al. (2015) Exp. Dermatology 24, 22-28). Here we have now tested whether TGF β , siLATS1/2 or WNT3a are able to induce invasive growth of proliferative melanoma cells. The results show that TGF β but not siLATS1/2 or WNT3a are able to promote invasive growth of the melanoma cells (**new Fig. 1c**).

- The poor effect of Wnt3a on the activation of mesenchymal genes and the requirement of b-catenin for the expression of TGF-b or TAZ-dependent genes suggest that the activation of these genes is not dependent of the canonical b-catenin-associated transcriptional factors Tcf4 or LEF1. This point is relevant and should be studied determining the effect of the depletion of these genes or using a dominant-negative form of Tcf4 or Lef1.

We agree with the reviewer on this important point. We have now performed siRNA-mediated ablation of the expression of TCF4 and LEF1 upon stimulation of proliferative melanoma cells with TGF β , siLATS1/2 or Wnt-3a. Knockdown of LEF1 Led to a reduced induction of Wnt target genes upon Wnt-3a treatment. However, knockdown of either LEF1 or TCF4 did not affect the induction of Wnt target genes by TGF β or LATS1/2. Also, the loss of the expression of melanocyte differentiation markers and the gain of the expression of mesenchymal markers by TGF β was not significantly affected by the siRNA-mediated ablation of TCF4 and LEF1. Finally, the siLATS1/2-induced expression of YAP/TAZ target genes was not affected by the knockdown of TCF4 and LEF1. These results have now been included into the manuscript as completely **new Suppl. Fig. 11**.

- The activation of Wnt-dependent genes by Wnt3a is lower than by TGFb. This is surprising and suggests that the Wnt3a pathway is constitutively activated in these cells. The authors should check this, at least analyzing b-catenin stabilization and translocation to the nucleus in the absence or presence of Wnt3a.

We appreciate the reviewer's comment and we believe that the quantitative differences in gene expression may be due to the complex interactions between the various signaling pathways activated by TGF β , i.e. that TGF β -dependent transcriptional activities cooperate with specific Wnt-3a/ β -catenin-mediated transcriptional activities, as can also be seen by the computational analysis of the RNA sequencing data.

To further assess the Wnt-3a- β -catenin-dependent activities induced by the different stimuli, we have analyzed nuclear localization of β -catenin by immunofluorescence microscopy analysis in two different proliferative melanoma cell lines. Compared to untreated cells (no nuclear β -catenin), Wnt-3a treatment increases the total amount of β -catenin and its nuclear localization. Treatment with TGF β also led to an increase of nuclear β -catenin, although to a lesser extent as observed with Wnt-3a. In contrast, treatment with siLATS1/2 did not apparently increase the nuclear localization of β -catenin (**new Suppl. Fig. 3**). These results indicate that β -catenin-mediated Wnt signaling is not constitutively active in proliferative melanoma cells. Consistent with this notion, siRNA-mediated ablation of TCF4 and LEF1 did not reduce the expression of canonical Wnt target genes in unstimulated proliferative melanoma cells (**new Suppl. Figure 11**, see also above).

- The results presented in Suppl Figs 4 and 5 do not provide any relevant information for the article and should be deleted, unless additional experiments are performed demonstrating that indeed b-catenin is interacting with N-cadherin. The title of this figures is that "b-catenin-mediated adhesion is not required for phenotype switching" whereas the results show that N-cadherin is not required for this switch. Moreover, these experiments present the problem that they have been performed with a not-specified siRNA that is not available anymore; so, they cannot be reproduced.

We appreciate the reviewer's comment. To demonstrate the interaction of β -catenin with N-cadherin we have now performed immunoprecipitation experiments which demonstrate a direct interaction of β -catenin with N-cadherin in proliferative cells that have been induced to undergo the mesenchymal phenotype switch either by TGF β or by siLATS1/2 (new Suppl. Fig. 9). Moreover, we have expanded the analysis of changes of gene expression of melanocytic, mesenchymal and Wnt/ β -catenin target genes upon siRNA-mediated depletion of N-cadherin expression (**new Suppl. Fig. 9b,c and new Suppl. Fig. 10a,b**). We have also clarified the sequence of the siRNA against N-cadherin. The sequence of the siRNA which had been custom made is now given in revised **Suppl. Table II**.

- The efficiency in the siRNA-mediated depletion of the different genes should be assessed by Western blot, that is more indicative than RT-PCR.

We have now performed immunoblotting analysis to demonstrate the efficiencies of the siRNA-mediated ablation of the expression of the genes of interest. These results are now shown in **new Suppl. Fig. 5c**.

- The authors need to provide a much better description of the cell lines used in these assays and in general, of the conditions used. Have the cell lines been authenticated?

The cell lines are patient-derived cell lines previously published and kindly provided by the colleagues of the University Hospital of Zürich who had previously established and published them. These cell lines are being widely used in the field, and we have now included a better description of their origin and their characterization in the Results section and in the Methods section.

- How confident are the authors that their siRNA do not present off-target effects? Only a combination of siRNAs has been used to deplete each target. At least some results (those obtained with b-catenin-depletion) should be rescued with a siRNA-resistant ectopic gene or validated using an alternative shRNA.

In addition to the siRNA pool from Dharmacon We have now repeated the siRNA-mediated ablation of β -catenin expression with additional siRNA sequences from Microsynth which reproduced the results. These direct comparisons are now shown in **new Suppl. Fig. 7a,b**.

- Other technical issues. What is the source of Wnt3a? Is it recombinant or coming from as conditioned-medium? Another b-catenin-target gene with a better stimulation by Wnt3a than NKD1 should be included in the Figure 1b analysis since this gene is not significantly upregulated by Wnt3a.

The source of Wnt-3a (commercially available, recombinant murine Wnt-3a from Preprotech) has now been specifically described in the Methods section. Besides NKD1, AXIN2 and NOTUM, we have now included CTLA4 as additional canonical WNT target gene (**revised Fig. 1b and Suppl. Fig. S2b; new Suppl. Fig. 4b,c; new Suppl. Fig. 7b; new Suppl. Fig. 10a,b; new Suppl. Fig. 11b**).

Reviewer #3

In the present ms. the authors analyze the hierarchical interaction of three signaling pathways, TGFb/SMAD, Hippo/YAP/TAZ, and Wnt/b-catenin, in the phenotypic switch of melanoma cells. By performing functional assays in two patient derived melanoma cells, combined with HTP sequencing and computational analyses (i.e, Nested Effect Models, NEM) they infer the hierarchical interaction among the mentioned pathways in the transition from the proliferative to the invasive phenotype. The main conclusion of the study is that transcriptional regulation by the three signaling pathways is required for the proliferative-to-invasive switch in melanoma. While TGFb/SMAD is on the top of the hierarchy followed by the Hippo pathway, and then by Wnt/b-catenin, b-catenin seems to play a dual role supporting both the proliferative phenotype and the invasive switch mediated by the other two pathways. This is an interesting study that provides novel information of basic mechanisms underlying the melanoma phenotypic switch and it can be of potential interest in the future for targeting specific pathways. The study is performed using state of the art experimental design and computational analyses in two melanoma cell lines. The results obtained after manipulating the signaling pathways in different combinations support the main conclusions of the authors on the hierarchical interaction of the pathways at transcriptional level. However, the main limitation of the study is the absence of biological studies to support that the transcriptional data indeed reflect distinct phenotypic states. At least some functional analyses of the effect of manipulation of the pathways on the phenotype should be included to increase the biological relevance of the present study. Other minor controls should be also considered before publication, as described below.

We thank the reviewer for her/his interests in our manuscript and more importantly, for the helpful and constructive comments to improve the quality of the manuscript.

Main points:

1. The "invasive" phenotype mediated by TGF β treatment, and to a lower extent by Hippo activation, is only determined at morphological level described as a more mesenchymal phenotype (F1g. 1a; Suppl Fig S2a) and EMT marker analyses by RT-qPCR (Fig. 1b; Suppl Fig. S2b). At least some functional invasion assay, i.e., in vitro assays on matrigel Boyden chambers, needs to be included to fully support the claimed phenotypic switch. The same applies to cells treated with Wnt3A in which the claim of "not effect in cell morphology" is not easily appreciated only from the images provided (F1g. 1a; Suppl Fig S2a).

We thank the reviewer for this suggestion which is similar to comment 1 by reviewer 2. We have now performed a modified Boyden chamber migration assay which demonstrates that TGF β but neither siLATS1/2 nor WNT3a are able to promote cell migration (**new Fig. 1d**).

In addition, we have performed 3D invasion assays in Matrigel culture conditions. These assays have shown increased invasive growth by the mesenchymal melanoma cells as compared to the proliferative melanoma cells, as reported previously by our laboratory (Schlegel et al. (2015) Exp. Dermatology 24, 22-28). Here we have now tested whether TGF β , siLATS1/2 or WNT3a are able to induce invasive growth of proliferative melanoma cells. The results show that TGF β but not siLATS1/2 or WNT3a are able to promote invasive growth of the melanoma cells (**new Fig. 1c**).

2. The strategy of using siCtrl melanoma cells to activate TGF β , Wnt and the Hippo pathways is correct to discard non-specific effects when using specific siRNAs (i.e., siLATS1/2). However, it should be important to show that similar phenotypic and transcriptomic effects are observed when parental melanoma cells are treated with the activating agents.

We thank the reviewer for this comment. To exclude any effects of siRNA transfection of the stimulation with TGF β and Wnt-3a we have now performed these experiments in the absence of any transfections with siCtrl. The results for the two proliferative (parental) melanoma cells lines show comparable results to the cells transfected with siCtrl at the level of changes in cell morphology and at the level of target gene expression. The results are highly comparable to the results from the experiments in the presence of siCtrl and they are now shown in **new Suppl. Fig. 4a,b** and **new Suppl. Fig. 6a,b**.

3. The RT-qPCR analysis of EMT-TFs expression in the different experimental situations should be extended to SLUG (SNAIL2), ZEB2 and TWIST1, at least after modulation of the individual pathways (Fig. 1b; Suppl Fig. S2b). As the expression of SNAIL2 and ZEB2 has been previously shown to be downregulated, in favor of ZEB1 and TWIST1, during the invasive switch in melanoma tumors by others (Caramel et al., Cancer Cell, 2013; Denecker

et al., Cell Death Differ, 2014), the authors should check if similar changes apply or not to their cellular models.

We thank the reviewer for this comment. The expression of SNAIL2, ZEB2 and TWIST1 was not significantly induced by the activation of the three pathways in two epithelial melanoma cell lines and hence we had not included these results in the initial submission. We have now included the results in the **new Suppl. Figure S2d**.

November 5, 2021

RE: Life Science Alliance Manuscript #LSA-2021-01010-TR

Prof. Gerhard Christofori
University of Basel
Dept. of Biomedicine
Mattenstrasse 28
Basel CH-4058
Switzerland

Dear Dr. Christofori,

Thank you for submitting your revised manuscript entitled "Hierarchy of TGF β /SMAD, Hippo/YAP/TAZ and Wnt/ β -catenin signaling in melanoma phenotype switching". We would be happy to publish your paper in Life Science Alliance pending final revisions necessary to meet our formatting guidelines.

- Please upload all figure files as individual ones, including the supplementary figure files
- please add Keywords for your manuscript in our system
- please add the Twitter handle of your host institute/organization as well as your own or/and one of the authors in our system
- please add Vincenzo Prestigiacomo to our system as contributing author and add his contribution to the system as well
- please make sure the author order in your manuscript and our system match
- please use the [10 author names, et al.] format in your references (i.e. limit the author names to the first 10)
- please use capital letters when introducing the panels in figure legends, in actual figures, and their callouts in the manuscript text
- please add your main, supplementary figure, and table legends to the main manuscript text after the references section
- please use Arabic numbers when labeling tables. Also, change their callouts in the manuscript text accordingly
- Please indicate molecular weight next to each protein blot
- Please provide the accession numbers for the RNA-seq data to the Data Availability Statement

A. FINAL FILES:

B. MANUSCRIPT ORGANIZATION AND FORMATTING:

Sincerely,

Reviewer #1 (Comments to the Authors (Required)):

All my comments have been addressed in the revised version of the manuscript, which helped to significantly improve the manuscript. I congratulate the authors for this nice piece of work and truly recommend this version for publication in Life Science Alliance.

Reviewer #2 (Comments to the Authors (Required)):

In this revised version the authors have adequately responded to my concerns. I think that the article is now suitable for publication.

Reviewer #3 (Comments to the Authors (Required)):

The authors have satisfactorily addressed the specific concerns raised on the original version. I congratulate them for the additional experimental work and clarifications that certainly have improved the quality of the ms.

Point-by-point reply to the editor's comments**Journal: Life Science Alliances****Manuscript: #LSA-2021-01010-TR****The hierarchy of TGF β /SMAD, Hippo/YAP/TAZ and Wnt/ β -catenin signaling in melanoma phenotype switching**

Fabiana Lüönd, Martin Pirkl, Mizue Hisano, Vincenzo Prestigiacomo, Ravi K.R. Kalathur, Niko Beerenwinkel, and Gerhard Christofori

Introductory Remarks

We have now revised the manuscript and the figures according to the recommendation by the editors.

The details of the revisions can be seen in the point-by-point reply to the reviewers' comments. We copied the reviewers' comments in *italic* and presented our reply in regular font.

Point-by-point reply

-Please upload all figure files as individual ones, including the supplementary figure files

Done!

-please add Keywords for your manuscript in our system

Done!

-please add the Twitter handle of your host institute/organization as well as your own or/and one of the authors in our system

The first and the corresponding last author do not have a Twitter account.

-please add Vincenzo Prestigiacomo to our system as contributing author and add his contribution to the system as well

Done!

-please make sure the author order in your manuscript and our system match

-please use the [10 author names, et al.] format in your references (i.e. limit the author names to the first 10)

Corrected and changed.

-please use capital letters when introducing the panels in figure legends, in actual figures, and their callouts in the manuscript text

Corrected and changed.

-please add your main, supplementary figure, and table legends to the main manuscript text after the references section

Done!

-please use Arabic numbers when labeling tables. Also, change their callouts in the manuscript text accordingly

Corrected and changed.

-Please indicate molecular weight next to each protein blot

Done!

-Please provide the accession numbers for the RNA-seq data to the Data Availability Statement

Done!

November 12, 2021

RE: Life Science Alliance Manuscript #LSA-2021-01010-TRR

Prof. Gerhard Christofori
University of Basel
Dept. of Biomedicine
Mattenstrasse 28
Basel CH-4058
Switzerland

Dear Dr. Christofori,

Thank you for submitting your Research Article entitled "Hierarchy of TGF β /SMAD, Hippo/YAP/TAZ and Wnt/ β -catenin signaling in melanoma phenotype switching". It is a pleasure to let you know that your manuscript is now accepted for publication in Life Science Alliance. Congratulations on this interesting work.

DISTRIBUTION OF MATERIALS:

Again, congratulations on a very nice paper. I hope you found the review process to be constructive and are pleased with how the manuscript was handled editorially. We look forward to future exciting submissions from your lab.

Sincerely,
